# Differences in Characteristics and Length of Stay of Elderly Emergency Patients before and after the Outbreak of COVID-19

**DOI:** 10.3390/ijerph20021162

**Published:** 2023-01-09

**Authors:** Wang-Chuan Juang, Sonia Ming-Jiu Chiou, Hsien-Chih Chen, Ying-Chun Li

**Affiliations:** 1Quality Management Center, Kaohsiung Veterans General Hospital, Kaohsiung 813414, Taiwan; 2Department of Business Management, National Sun Yat-sen University, Kaohsiung 804201, Taiwan; 3Institute of Health Care Management, National Sun Yat-sen University, Kaohsiung 804201, Taiwan

**Keywords:** COVID-19, elderly, emergency care, length of stay

## Abstract

(1) Background: COVID-19 has spread worldwide and affected Taiwan’s medical system and people’s lives. This study aimed to explore the impact of medical utilization on the characteristics and length of stay (LOS) of elderly emergency department (ED) patients before and after COVID-19; (2) Methods: We gathered ED visits from January to September 2019 (pre-pandemic group) and from January to September 2020 (pandemic group). The data analysis methods included descriptive statistics, the Pearson’s chi-square test, the independent sample *t*-test, and binary logistic regression; (3) Results: In 2020, during COVID-19, a significant decrease in ED monthly visits occurred from January; the maximum decrease was 32% in March. The average LOS during COVID-19 was shortened, with a significant reduction in diagnoses compared with the pre-pandemic period; (4) Conclusions: The threat of COVID-19 has changed the elderly’s behavior in ED visits and shortened the LOS of ED. The study’s results emphasize the importance of analyzing the medical utilization of elderly ED patients and understanding the medical quality of healthcare institutions. With Taiwan’s rapidly aging society, the demand for healthcare increases from time to time. The overcrowding of medical attention is often a problem. The results recommend that the overcrowding problem has the opportunity to be solved.

## 1. Introduction

Since December 2019, Coronavirus Disease 2019 (COVID-19), caused by Severe Acute Respiratory Syndrome Coronavirus 2 (SARS-COV-2), has spread worldwide. With the increasing number of confirmed cases and deaths, the World Health Organization declared a pandemic on 11 March 2020 [1]. Since the outbreak, COVID-19 has had a serious impact on medical and health systems, people’s health, and lives worldwide.

During this study, there was no effective vaccine for prevention and specific treatment methods [2], so the emergency department (ED) plays an important role. When threatened by COVID-19, EDs must take appropriate infection control procedures and contingency plans to manage patients with suspected COVID-19 [3]. Many patients feel anxious and confused about the uncertainty of the virus and the risk of infection; this changes their willingness to seek emergency care, leading to increased morbidity and mortality [4,5]. Several studies have shown a decrease in ED visits due to the COVID-19 epidemic [6,7,8,9,10,11,12,13]. In Taiwan, the government learned from the SARS experience in 2003, quickly implemented the epidemic prevention mechanism, and adopted new technologies and big data analytics for COVID-19 [14]. Compared with other countries, the epidemic situation in Taiwan is relatively stable, but COVID-19 still has an impact on EDs. A study of a medical center in Taiwan pointed out that the total number of ED visits showed a decreasing trend [15].

Studies have found that the elderly are more susceptible to COVID-19, with highly severe cases and high case fatality [2,16,17], and the case fatality rate increases with age [18]. According to a report from the U.S. Centers for Disease Control and Prevention (CDC), elderly patients account for 31% of the total number of COVID-19 patients, 45% of hospitalized patients, 53% of ICU patients, and 80% of deaths [19]. Elderly patients are more likely to suffer from multiple chronic diseases and comorbidity [20,21]. Studies have pointed out that elderly patients with COVID-19 are often associated with cardiovascular disease, diabetes, hypertension, COPD (Chronic obstructive pulmonary disease), and other serious chronic diseases [17,18]. The fatality rate and severity of the disease are related to age and comorbidity [16].

In recent years, Taiwan and other countries have devoted themselves to exploring the status of elderly patients in EDs. Several Taiwanese studies showed that elderly patients are frequent ED users and stay longer because their conditions are usually more serious and complex, with nonspecific symptoms [21,22]. Due to the rapidly aging population, Taiwan’s elderly patients have become a great burden and occupy emergency medical resources, so it is necessary to understand the utilization of emergency medical care by elderly patients.

In Taiwan, limited COVID-19 studies investigated the use of emergency medical services. By observing boarded patients who were staying in the ED waiting to be transferred to the general ward, the study aimed to explore the difference in characteristics and length of stay (LOS) of elderly emergency department (ED) patients and boarded patients over 65 years of age before and after the COVID-19 outbreak and to understand the impact of medical utilization caused by the COVID-19 outbreak.

## 2. Materials and Methods

### 2.1. Study Design and Data Collection

This study was a quantitative, descriptive-correlational, and longitudinal study using retrospective data from Kaohsiung Veterans General Hospital, a major public medical center in southern Taiwan. After the data collection, we excluded ED patients < 64 years old and missing data according to the study design. These subjects were divided into two groups: the COVID-19 pandemic period from January 2020 to September 2020 and the pre-pandemic period from January 2019 to September 2019.

The study aimed to examine whether the difference and impact of ED use in elderly patients before and after the COVID-19 outbreak were statistically significant. The independent variables included demographic variables (sex, age group), triage (1, 2, 3, 4, 5), after the movement of emergency disposition (discharge, transfer to the general ward, transfer to intensive care unit (ICU)), against advice discharge (AAD), died in ED, transfer to other hospitals, self-discharge, bed request (internal medicine, surgery, others), and length of ED stay (LOS). Bed requests in this study referred to boarded patients who had to be transferred to the general ward, which could be divided into internal medicine, surgery, and other categories.

The Taiwan Triage and Acuity Scale (TTAS) was used for our triage category. There are five triage categories ranked by level of acuity: category 1, resuscitation; category 2, emergency; category 3, urgent; category 4, less urgent; and category 5, non-urgent [23].

### 2.2. Statistical Analysis

These analyses were performed using IBM SPSS version 24.0 (SPSS Inc., Chicago, IL, USA). The data analysis uses descriptive statistics, the Pearson’s chi-square test, the independent sample *t*-test, and binary logistic regression. The frequency distribution and percentage were used to describe the distribution of the variables. The Pearson’s chi-square test was used to analyze the distribution difference between each variable’s proportion before and after the COVID-19 outbreak. An independent sample *t*-test was used to analyze the difference in emergency treatment outcomes (non-hospitalization, hospitalization, transfer to the general ward, transfer to ICU) and bed request before and after the COVID-19 epidemic. Multivariate logistic regression was used to determine the relationship between each variable and ED LOS 24 h. Odds ratios (OR) and 95% confidence intervals (CI) were calculated. A *p*-value < 0.05 was considered statistically significant. Ethical approval was provided by the Institutional Review Board of Kaohsiung Veterans General Hospital (IRB No.: KSVGH21-CT4-11).

## 3. Result

A total of 41,786 ED patients visited between January and September 2019 and 35,216 between January and September 2020, and elderly patients (aged over 65) accounted for 30,073 (39.1%). The percentage of elderly visits was 39.4% in 2019 pre-pandemic and 38.7% in the 2020 COVID-19 pandemic.

Table 1 shows the characteristics of elderly patients before and after the COVID-19 outbreak. There were no significant differences in sex distribution, age group, or triage. However, disposition before and after the COVID-19 outbreak showed statistically significant differences. Amid the COVID-19 pandemic, the proportion of patients allowed to discharge (55.7% vs. 50.3%) was lower than in the pre-pandemic period. In comparison, the proportion of patients transferred to the general ward (35.3% vs. 40.1%) and the ICU (3.3% vs. 3.8%) and who died in ED (0.8% vs. 1.3%) was higher than in the pre-pandemic period. For elderly boarded patients, analysis was conducted according to bed request. There was no significant difference in bed request between the two groups (Table 2).

In 2020, a significant decrease in ED monthly visits occurred from January to March. However, after April the number of ED visits gradually increased. The largest number of patients (*n* = 1856) was in January, and the lowest (*n* = 1247) was in March (Figure 1A). The maximum decrease in the percentage of monthly visits during the COVID-19 outbreak compared to the pre-pandemic period was 32% in March. Monthly visits gradually decreased and reached around 10% from June to August; however, there was a higher decreasing trend in September (Figure 1B).

Regarding treatment outcome, there was a decrease in the overall of average ED LOS during the COVID-19 pandemic period, with 6.7 ± 10.9 h for non-hospitalization patients, 18.0 ± 16.6 for hospitalization patients, 18.4 ± 16.6 for patients transferred to the general ward, and 14.0 ± 16.5 for patients transferred to ICU. Non-hospitalization, hospitalization, and transfer to the general ward were significantly lower than in the pre-pandemic period in LOS (*p* < 0.001).

Regarding bed request, the average ED LOS decreased in internal medicine, 19.1 ± 16.9; surgery, 13.4 ± 12.5; and others, 10.2 ± 12.0 after the COVID-19 outbreak. In addition, internal medicine and surgery showed significant differences in LOS before and after COVID-19 (*p* < 0.05) (Table 3).

Table 4 shows the difference in the ED diagnosis and the average LOS during the COVID-19 pandemic period and the pre-pandemic period. During the COVID-19 outbreak, there was a significant reduction in the number of diagnoses apart from neoplasms. For the pre-pandemic period, the top five average ED LOS by diagnosis were infectious and parasitic diseases with 25.6 ± 26.3, diseases of the respiratory system with 21.1 ± 24.9, diseases of the skin and subcutaneous tissue with 20.7 ± 29.6, diseases of the genitourinary system with 20.4 ± 26.3, and diseases of the digestive system with 18.6 ± 21.8. For the COVID-19 pandemic period, the top five average ED LOS by diagnosis were diseases of the respiratory system with 15.5 ± 17.3, infectious and parasitic diseases with 14.9 ± 14.1, diseases of the genitourinary system with 14.8 ± 16.2, diseases of the digestive system with 13.7 ± 15.6, and diseases of the circulatory system with 12.8 ± 16.6. There was a significant difference in ED LOS by diagnosis, apart from musculoskeletal system and connective tissue diseases.

The results of the logistic regression analysis for EDLOS ≥ 24 h are presented in Table 5. For the pre-pandemic period, having female as a reference category, then male’s OR = 1.180, *p* < 0.001, increased the probability of EDLOS ≥ 24 h. Having age 65–74 patients as a reference category, then age 75–84 patients’ OR = 1.119, *p* = 0.027 and ≥85 patients’ OR = 1.360, *p* < 0.001, increased the probability of EDLOS ≥ 24 h. Having triage 4 patients as a reference category, then triage 3 patients’ OR = 2.040, *p* = 0.010, increased the probability of EDLOS ≥ 24 h. Having non-hospitalization patients as a reference category, then internal medicine patients’ OR = 10.376, *p* < 0.001 and surgical patients’ OR = 2.008, *p* < 0.001, increased the probability of EDLOS ≥ 24 h.

During the COVID-19 pandemic period, having female patients as a reference category, then male patients’ OR = 1.146, *p* = 0.012, increased the probability of EDLOS ≥ 24 h. Having age 65–74 patients as a reference category, then age 75–84 patients’ OR = 1.113, *p* = 0.045, increased the probability of EDLOS ≥ 24 h. Having non-hospitalization patients as a reference category, then internal medicine patients’ OR = 7.065, *p* < 0.001 and surgical patients’ OR = 2.872, *p* < 0.001, increased the probability of EDLOS ≥ 24 h.

## 4. Discussion

In this study, elderly patients accounted for 39.1% of ED utilization rates; the proportion was significantly higher than in the previous study. Compared with the study by Chou et al., elderly patients account for 12–24% of ED visits [22]. This may be due to the impact of the medical system; most of the patients at the hospital in southern Taiwan used for our study are elderly, including veterans and their families. In addition, this hospital has a geriatric medicine building, reflecting the hospital’s attention to elderly patients. So the proportion of elderly patients in the ED would naturally be higher.

Compared to the pre-pandemic period, the number of ED monthly visits was unaffected by the COVID-19 pandemic, which did not show any particular trend. The possible reason is that elderly patients often have multiple chronic diseases and more nonspecific and complex clinical symptoms or signs. Therefore, they still need to receive treatment, if necessary, even during the COVID-19 outbreak [21,22].

According to the analysis before and after the outbreak of COVID-19, there were no statistically significant differences in the triage of elderly patients in the ED. But several studies have demonstrated that the proportion of non-urgent category 4 and 5 patients had reduced during the SARS or during the COVID-19 outbreak [10,24]. However, our finding reflected that both mild or severe elderly patients were not affected by COVID-19; a possible explanation for this result is that many elderly patients with mild respiratory symptoms visited the ED for screening due to concern about COVID-19 infection; in addition, elderly patients more frequently have a severer triage category.

During the COVID-19 pandemic, the percentage of elderly patients allowed to discharge decreased, while the percentage of elderly patients transferred to the general ward and the ICU increased. This finding is consistent with the results of another study which found that patients had a higher demand for inpatient care following the COVID-19 outbreak [25]. Notably, the percentage of elderly patients who died in ED was higher than in the pre-pandemic period. This could be interpreted as that the COVID-19 outbreak caused panic and the delaying of treatment for people who might need it, thereby endangering their life safety [4,5].

The average ED LOS decreased in inpatients and outpatients affected by the COVID-19 outbreak. This finding might reflect that most patients were afraid of staying in the hospital, leading to more available hospital beds and thus reducing the waiting time for admission. However, patients transferred to the ICU showed no significant difference in average ED LOS. This likely reflects that elderly patients usually have severe symptoms and still require aggressive treatment in the ICU [21,22,26]. Regarding elderly boarded patients, this study found that internal medicine and surgery significantly decreased in the average ED LOS. In contrast, there was no significant difference from the others. The possible reason for this is twofold: the clinical manifestations of some elderly patients require more time to run some medical tests so as to understand the changes in the condition, and more check-ups are needed to find out the cause of the disease. In the pre-pandemic period, each patient, whether elderly or not, required more time to be admitted to the internal medicine department, which prolongs the average LOS; however, during the COVID-19 pandemic, the decrease in non-elderly patients led to elderly patients getting prompter attention in ED.

We further investigated the diagnosis of the disease and found a decline in the number of ED visits and in average ED LOS for respiratory and digestive diseases. However, another study in the United States stated that the diagnoses associated with lower respiratory disease, difficulty breathing, and pneumonia increased during the COVID-19 pandemic [7]. This finding reflected that the epidemic in Taiwan was relatively stable during the study period and not as serious as in other countries. In addition, wearing masks and washing hands frequently may also reduce the incidence of respiratory tract infection, gastroenteritis, and other diseases.

After the outbreak of the COVID-19 pandemic, elderly patients who were male, aged 75–84, and underwent internal medicine or surgery had a higher odds ratio for ED LOS over 24 h. Nevertheless, male gender was still a significant factor compared with the pre-pandemic period, and the odds decreased after the pandemic. However, patients aged over 85 and patients with triage category 3 had a higher odds ratio after the outbreak, although it did not reach statistical significance. Regarding bed request, the odds ratio of internal medicine decreased significantly, while surgery increased significantly. This implies that these elderly patients have more serious medical conditions and need hospitalization for emergency operations.

This study has some limitations. First, we described ED service used in a single medical center in southern Taiwan; the results of this study were not representative of the utilization of emergency medical service resources in other hospitals in Taiwan. Secondly, due to time limitations, we cannot predict the impact of a longer COVID-19 pandemic on further changes in ED visits. Third, during the study period, the change in the intention of ED visits in our study might not have been caused by COVID-19 or might have been caused by other factors, so we cannot directly understand the elderly patients’ real behavioral motivation for seeking care.

## 5. Conclusions

In the study, COVID-19 didn’t affect ED visits of elderly patients or show large differences in bed requests, but it did significantly impact length of stay. This finding suggests that ED crowding problems could have solutions, because the main reason for ED crowding comes from serious long-term stay. Therefore, the COVID-19 pandemic could be regarded as an intervention, and adequate hospital ED admission policy and regulations could be an intervention as well and could solve the ED crowding problem. The study’s results emphasize the importance of analyzing the medical utilization of elderly ED patients worldwide. With Taiwan’s rapidly aging society, it is recommended that medical institutions improve medical quality and patient safety by understanding the complex and nonspecific symptoms of elderly patients in the emergency room. In addition, the current COVID-19 outbreak in Taiwan is still active and ongoing. Therefore, we suggest updating the study sample to analyze changes in long-term trends.

## Figures and Tables

**Figure 1 ijerph-20-01162-f001:**
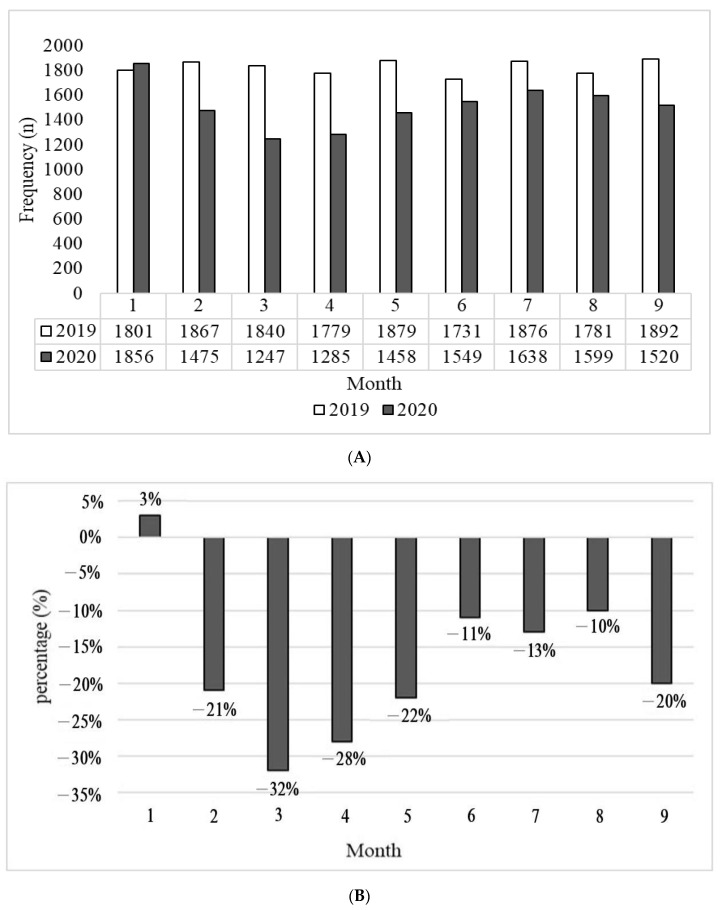
(**A**) Total number of emergency department (ED) visits between the COVID-19 pandemic and the pre-pandemic period. (**B**) Comparison of percentage changes in emergency department (ED) visits between the COVID-19 pandemic and the pre-pandemic period.

**Table 1 ijerph-20-01162-t001:** Comparison of the characteristics of elderly patients in emergency department (ED) during the COVID-19 pandemic versus during the pre-pandemic period.

	Before COVID-19	During COVID-19	*p*-Value
	(*n* = 16,446)	(*n* = 13,627)	
	*n* (%)	*n* (%)	
Sex			0.391
Male	9058 (55.1)	7438 (54.6)	
Female	7388 (44.9)	6189 (45.4)	
Age (years)			
65–74	6902 (42.0)	5828 (42.8)	0.343
75–84	5325 (32.4)	4326 (31.7)	
≥85	4219 (25.7)	3473 (25.5)	
Triage			0.138
1	1286 (7.8)	1107 (8.1)	
2	4238 (25.8)	3570 (26.2)	
3	10,673 (64.9)	8702 (63.9)	
4	226 (1.4)	223 (1.6)	
5	23 (0.1)	25 (0.2)	
Disposition			0.000
Discharge	9156 (55.7)	6851 (50.3)	
Transfer to the general ward	5835 (35.5)	5461 (40.1)	
Against advice discharge	658 (4.0)	541 (4.0)	
Transfer to intensive care unit	546 (3.3)	520 (3.8)	
Transfer to other hospitals	91 (0.6)	65 (0.5)	
Self-discharge	26 (0.2)	9 (0.1)	
Died in ED	134 (0.8)	180 (1.3)	

**Table 2 ijerph-20-01162-t002:** A comparison of the characteristics of elderly boarded patients in the emergency department (ED) during the COVID-19 pandemic versus during the pre-pandemic period.

	Before COVID-19	During COVID-19	*p*-Value
	(*n* = 5835)	(*n* = 5461)	
	*n* (%)	*n* (%)	
Bed request			0.295
Internal medicine	5264 (90.2)	4878 (89.3)	
Surgery	471 (8.1)	482 (8.8)	
Others	100 (1.7)	101 (1.8)	

**Table 3 ijerph-20-01162-t003:** Comparison of elderly patients’ and boarded patients’ length of stay in the emergency department (ED) during the COVID-19 pandemic versus during the pre-pandemic period.

	Before COVID-19		During COVID-19		*p*-Value
	*n*	Mean ± SD (h)	*n*	Mean ± SD (h)	
Emergency treatment outcome					
Non-hospitalization	10,065	9.1 ± 16.0	7646	6.7 ± 10.9	0.000
Hospitalization	6381	27.4 ± 25.9	5981	18.0 ± 16.6	0.000
Transfer to the general ward	5835	28.5 ± 26.2	5461	18.4 ± 16.6	0.000
Transfer to intensive care unit	546	15.6 ± 18.7	520	14.0 ± 16.5	0.137
Bed request					
Internal medicine	5264	30.0 ± 26.7	4878	19.1 ± 16.9	0.000
Surgery	471	15.4 ± 16.0	482	13.4 ± 12.5	0.026
Others	100	10.3 ± 10.1	101	10.2 ± 12.0	0.942

**Table 4 ijerph-20-01162-t004:** Primary medical diagnoses (International Classification of Disease, 10th revision) of elderly patients and the length of stay in the emergency department (ED).

Before COVID-19 (*n* = 16,446)			During COVID-19 (*n* = 13,627)			*p*-Value
Diagnosis	*n* (%)	Mean ± SD (h)	Diagnosis	*n* (%)	Mean ± SD (h)	
Symptoms, signs and abnormal clinical and laboratory findings not elsewhere classified	5710 (34.7)	13.5 ± 19.7	Symptoms, signs and abnormal clinical and laboratory findings not elsewhere classified	4097 (30.1)	9.6 ± 12.9	0.000
Neoplasms	1716 (10.4)	15.4 ± 20.4	Neoplasms	1846 (13.5)	12.0 ± 14.3	0.000
Diseases of the circulatory system	1656 (10.1)	16.5 ± 21.8	Diseases of the circulatory system	1546 (11.3)	12.8 ± 16.6	0.000
Diseases of the genitourinary system	1644 (10.0)	20.4 ± 26.3	Diseases of the genitourinary system	1380 (10.1)	14.8 ± 16.2	0.000
Diseases of the respiratory system	1364 (8.3)	21.1 ± 24.9	Diseases of the respiratory system	1050 (7.7)	15.5 ± 17.3	0.000
Diseases of the digestive system	1266 (7.7)	18.6 ± 21.8	Diseases of the digestive system	852 (6.3)	13.7 ± 15.6	0.000
Certain infectious and parasitic diseases	611 (3.7)	25.6 ± 26.3	Certain infectious and parasitic diseases	446 (3.3)	14.9 ± 14.1	0.000
Diseases of the musculoskeletal system and connective tissue	550 (3.3)	9.1 ± 16.9	Diseases of the musculoskeletal system and connective tissue	426 (3.1)	8.0 ± 12.8	0.279
Diseases of the skin and subcutaneous tissue	466 (2.8)	20.7 ± 29.6	Diseases of the skin and subcutaneous tissue	409 (3.0)	12.5 ± 23.0	0.000
Diseases of the nervous system	462 (2.8)	8.8 ± 21.7	Diseases of the nervous system	262 (1.9)	8.9 ± 12.0	0.004

**Table 5 ijerph-20-01162-t005:** Logistic model regression results for the predictors of EDLOS ≥ 24 h comparing the COVID-19 pandemic to the pre-pandemic period.

	Before COVID-19				During COVID-19			
	*p*-Value	OR	95% CI for OR		*p*-Value	OR	95% CI for OR	
			Lower	Upper			Lower	Upper
Sex (ref: Female)								
Male	0.000	1.180 ***	1.084	1.285	0.012	1.146 *	1.031	1.273
Age (years) (ref: 65–74)								
75–84	0.027	1.119 *	1.013	1.235	0.045	1.133 *	1.003	1.281
≥85	0.000	1.360 ***	1.227	1.507	0.312	1.069	0.940	1.216
Triage (ref: 4)								
1	0.149	0.661	0.377	1.160	0.769	1.102	0.578	2.099
2	0.380	1.278	0.739	2.212	0.637	1.164	0.619	2.189
3	0.010	2.040 **	1.185	3.512	0.157	1.570	0.840	2.935
Bed request (ref: Non-hospitalization)								
Internal medicine	0.000	10.376 ***	9.462	11.378	0.000	7.065 ***	6.246	7.990
Surgery	0.000	2.088 ***	1.663	2.622	0.000	2.872 ***	2.220	3.715
Others	0.258	1.470	0.754	2.867	0.762	0.836	0.262	2.671

(* *p* < 0.05; ** *p* < 0.01; *** *p* < 0.001).

## Data Availability

Research data will be available through reasonable request and get permission from the Kaohsiung Veterans General Hospital.

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
