# Peer review of "Differences in Characteristics and Length of Stay of Elderly Emergency Patients before and after the Outbreak of COVID-19"

_ijerph, 2023, doi:10.3390/ijerph20021162_

Round 1

Reviewer 1 Report

I am grateful for the opportunity to review this article, as this is a population I am concerned about and the target of my investigation.

Summary:

The method should emphasize the type of study in a more objective way (Quantitative, descriptive-correlational and longitudinal study). Regarding the results and conclusions, the authors present them adequately and easily perceptible by the readers.

Introduction:

The authors present a brief review of the problem under study, pointing out the most relevant aspects for its understanding, as well as an adjusted and current bibliography on the subject.

Material and methods:

Adequately described and understandable by readers. I think that line n.23 is shown incorrectly.

Results:

The presentation of results is adequate and understandable for the readers. No inconsistencies or errors in their presentation were identified. They respond to the lineado objective.

Figure 1 shows the term "month", however, the months are not shown.

Discussion:

The authors confront their results with studies that corroborate and do not corroborate their results, presenting the main reasons why this may not occur. They use adjusted bibliography to discuss the results.

Conclusion:

The authors respond to the objective of the study.

Congratulations and wishes for many successes on behalf of older people.

Reviewer 2 Report

(99-100) Please present the total number of ED visit for each period- pre-pandemic and pandemic and the percent of elderly patients for each.

In Discussion, I consider  important to present the in- hospital protocols dedicated to COVID-19 patients- decreasing the number of beds, limitation in surgical non-urgent intervention,...

In results - present the number of patients included in the study with COVID 19 pathology and the percent of ICU COVID 19 admitted patients

Reviewer 3 Report

Thank you for the opportunity to review your manuscript. The hypothesis tested is very relevant as you have outlined. The context of Taiwan is also explained well, given the variable approaches and responses to the pandemic worldwide. There were a few areas that i thought could be improved to enhance readership engagement.

There are minor grammatical errors throughout. Please check. 

Your title doesn't match what your main theme in the manuscript is, that is, you were preferentially looking at ED presentations of elderly, and a secondary outcome was LOS. An example of how this is true is looking at the conclusion - not one mention of LOS. 

Your discussion of the unchanged presentations pre and post pandemic is interesting. Another point that is worth exploring is the community services available in Taiwan - primary care, community support. If the elderly are well supported in the community, only sick elderly patients will present regardless of the pandemic. I suspect this is testament to a good community system in Taiwan, but may be wrong.
